# Structural Basis of Peptide-Based Antimicrobial Inhibition of a Resistance-Nodulation-Cell Division Multidrug Efflux Pump

Meinan Lyu,[a] Julio C. Ayala,[b,c] Isabella Chirakos,[a] Chih-Chia Su,[a] William M. Shafer,[b,c,d] Edward W. Yu[a]

[a]Department of Pharmacology, Case Western Reserve University School of Medicine, Cleveland, Ohio, USA
[b]Department of Microbiology and Immunology, Emory University School of Medicine, Atlanta, Georgia, USA
[c]Emory Antibiotic Resistance Center, Emory University School of Medicine, Atlanta, Georgia, USA
[d]Laboratories of Microbial Pathogenesis, VA Medical Center, Decatur, Georgia, USA

**ABSTRACT** Bacterial efflux pumps in the resistance-nodulation-cell division (RND) family of Gram-negative bacteria contribute significantly to the development of antimicrobial resistance by many pathogens. In this study, we selected the MtrD transporter protein of *Neisseria gonorrhoeae* as it is the sole RND pump possessed by this strictly human pathogen and can export multiple antimicrobials, including antibiotics, bile salts, detergents, dyes, and antimicrobial peptides. Using knowledge from our previously published structures of MtrD in the presence or absence of bound antibiotics as a model and the known ability of MtrCDE to export cationic antimicrobial peptides, we hypothesized that cationic peptides could be accommodated within MtrD binding sites. Furthermore, we thought that MtrD-bound peptides lacking antibacterial action could sensitize bacteria to an antibiotic normally exported by the MtrCDE efflux pump or other similar RND-type pumps possessed by different Gram-negative bacteria. We now report the identification of a novel nonantimicrobial cyclic cationic antimicrobial peptide, which we termed CASP (cationic antibiotic-sensitizing peptide). By single-particle cryo-electron microscopy, we found that CASP binds within the periplasmic cleft region of MtrD using overlapping and distinct amino acid contact sites that interact with another cyclic peptide (colistin) or a linear human cationic antimicrobial peptide derived from human LL-37. While CASP could not sensitize *Neisseria gonorrhoeae* to an antibiotic (novobiocin) that is a substrate for RND pumps, it could do so against multiple Gram-negative, rod-shaped bacteria. We propose that CASP (or future derivatives) could serve as an adjuvant for the antibiotic treatment of certain Gram-negative infections previously thwarted by RND transporters.

**IMPORTANCE** RND efflux pumps can export numerous antimicrobials that enter Gram-negative bacteria, and their action can reduce the efficacy of antibiotics and provide decreased susceptibility to various host antimicrobials. Here, we identified a cationic antibiotic-sensitizing peptide (CASP) that binds within the periplasmic cleft of an RND transporter protein (MtrD) produced by *Neisseria gonorrhoeae*. Surprisingly, CASP was able to render rod-shaped Gram-negative bacteria, but not gonococci, susceptible to an antibiotic that is a substrate for the gonococcal MtrCDE efflux pump. CASP (or its future derivatives) could be used as an adjuvant to treat infections for which RND efflux contributes to multidrug resistance.

**KEYWORDS** MtrD, *Neisseria gonorrhoeae*, membrane proteins, multidrug efflux, multidrug resistance, secondary transporter mechanism

Address correspondence to Edward W. Yu, edward.w.yu@case.edu.
The authors declare no conflict of interest.

**M**ultidrug efflux is considered to be a major cause of the failure of drug-based treatments of infectious diseases, and this property appears to be increasing in prevalence (1). Bacterial multidrug efflux pumps can have enormous clinical consequences as their expression can provide bacteria with the ability to resist many clinically relevant

antibiotics (2). In *Neisseria gonorrhoeae*, the best-characterized and most clinically important multidrug efflux system is the multiple transferrable resistance (MtrCDE) tripartite efflux pump (3–10), a member of the widely distributed resistance-nodulation-cell division (RND) superfamily in Gram-negative bacteria. This superfamily of multidrug efflux systems is capable of contributing to resistance to a variety of antimicrobial agents, including macrolides, $\beta$-lactams, cationic antimicrobial peptides, bile salts, dyes, and detergents (11). The *mtrCDE* locus of *N. gonorrhoeae* consists of three transcriptionally linked genes encoding the MtrC, MtrD, and MtrE efflux proteins, where all three components are required for mediating antibiotic resistance. *N. gonorrhoeae* MtrD contains multidrug binding sites and a proton relay network that generates the proton motive force (PMF) essential for drug expulsion from the bacterial cell. This multidrug efflux pump couples with the MtrC periplasmic membrane fusion protein and the MtrE outer membrane channel to actively extrude antimicrobials and mediate drug resistance (3–10). It has been shown that the overexpression of MtrCDE significantly contributes to clinically relevant levels of resistance to $\beta$-lactams and macrolides (11). In addition, the MtrCDE tripartite efflux system is capable of enhancing long-term colonization of the mouse vaginal mucosal layer, whereas gonococci lacking this efflux system are highly attenuated (3, 8, 12).

To elucidate the molecular mechanisms underlying drug recognition and extrusion via the MtrD multidrug efflux pump, we previously determined the crystal structure of MtrD$_{PID332}$ (MtrD from *N. gonorrhoeae* strain PID332) (9). We also recently solved single-particle cryo-electron microscopy (cryo-EM) structures of MtrD$_{CR103}$ (MtrD from the gonococcal strain CR.103) (13) carrying a mosaic-like sequence, which is capable of enhancing the activity of the pump and further elevating the levels of resistance to several antimicrobials, including macrolides (e.g., erythromycin and azithromycin) (11). These structures allowed us to identify the efflux pump's multidrug binding sites and modes of drug binding. The structural information also enabled us to understand how the proton relay network, which generates the PMF, couples with and facilitates drug extrusion by the pump.

With this structural knowledge in mind, we are interested in designing peptide-based inhibitors to hinder the function of multidrug RND efflux pumps. In particular, we have sought to discover novel peptide-based antimicrobial adjuvants that have the potential to allow the return of antibiotics previously withdrawn from treatment regimens. Here, we identify a cyclic cationic antibiotic-sensitizing peptide (CASP) that interacts with the MtrD multidrug efflux pump using amino acid contact sites distinct from those of two other cationic peptides that possess antibacterial activity but can be exported by MtrD (14–17). Critically, we found that CASP could render certain Gram-negative bacteria other than *N. gonorrhoeae* susceptible to an RND antibiotic substrate (novobiocin [Nov]), suggesting that it could serve as an adjuvant for antibiotic treatment regimens that have been thwarted by multidrug RND efflux pump systems similar to the gonococcal MtrCDE pump.

## RESULTS

**Screening of anti-MtrD heptapeptides.** Recently, peptide-based inhibitors have been found to be a promising therapeutic approach to the treatment of bacterial, viral, and malaria infections (18–31). To explore the possibility of designing peptide-based inhibitors as "antimicrobial adjuvants" specific for RND multidrug efflux pumps, we chose the PhD-C7C phage display combinatorial peptide library (32–37). Biopanning of this peptide library was used to select heptapeptides that specifically bind to the purified *N. gonorrhoeae* MtrD protein (strain FA19). Each heptapeptide in the library is flanked by a pair of cysteines located at the N and C termini. Under nonreducing conditions, the cysteines spontaneously form a disulfide cross-link, resulting in the phage display of cyclized peptides suitable for peptide-based therapeutics (38, 39). After three rounds of successive screening procedures, six novel anti-MtrD peptides (ADP1 to ADP6) capable of specifically binding to the MtrD protein were identified (see Table S1 in the supplemental material).

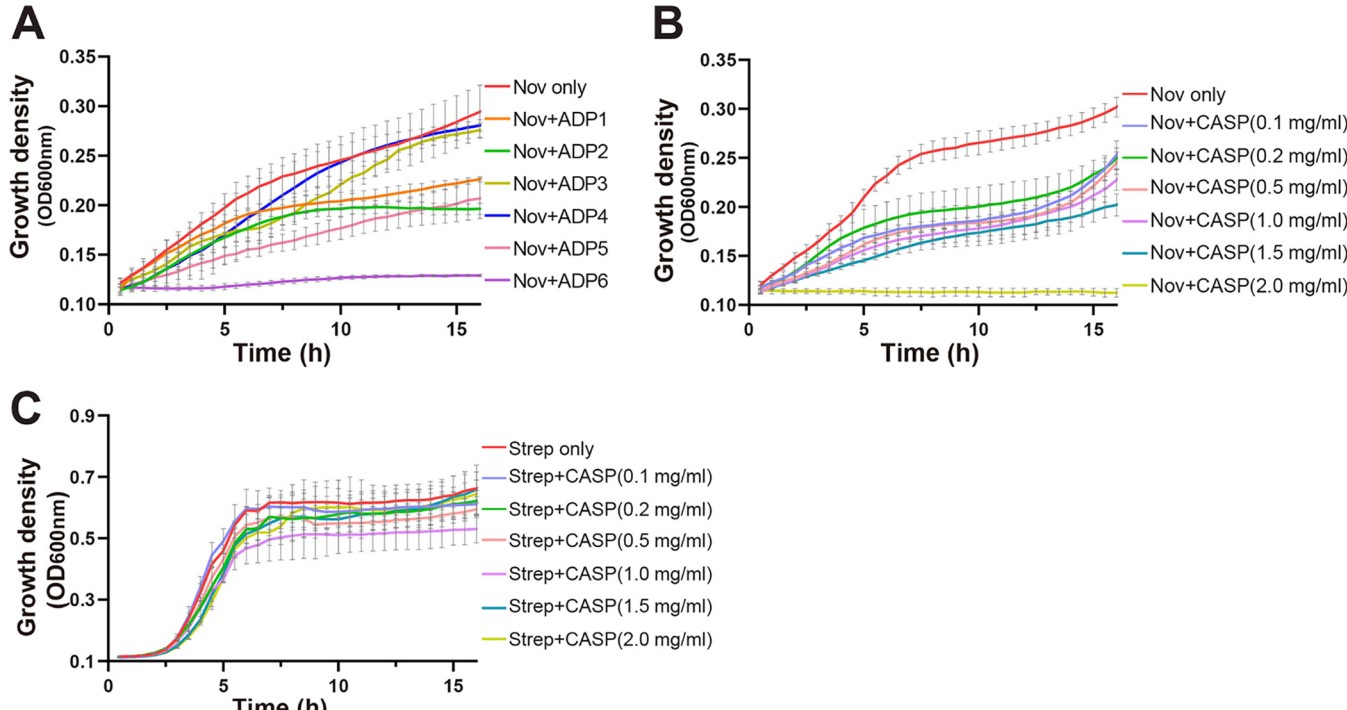

**FIG 1** Growth of *E. coli* BL21(DE3) Δ*acrB*/pACYCΩ*mtrCDE* cells. (A) Growth of cells in the presence of 75 μg/mL of Nov and supplemented with 2.0 mg/mL of ADP1 (34.4% inhibition compared with cell growth in the presence of 75 μg/mL of Nov only at the 15-h time point [*P* < 0.0001 by Student's *t* test]), ADP2 (49.8% inhibition compared with cell growth in the presence of 75 μg/mL of Nov only at the 15-h time point [*P* < 0.0001 by Student's *t* test]), ADP3 (6.5% inhibition compared with cell growth in the presence of 75 μg/mL of Nov only at the 15-h time point [*P* = 0.6173 by Student's *t* test]), ADP4 (1.2% inhibition compared with cell growth in the presence of 75 μg/mL of Nov only at the 15-h time point [*P* = 0.9997 by Student's *t* test]), ADP5 (47.8% inhibition compared with cell growth in the presence of 75 μg/mL of Nov only at the 15-h time point [*P* < 0.0001 by Student's *t* test]), or ADP6 (92.1% inhibition compared with cell growth in the presence of 75 μg/mL of Nov only at the 15-h time point [*P* < 0.0001 by Student's *t* test]). The growth curve with 75 μg/mL of Nov only is presented. (B) Growth of cells in the presence of 75 μg/mL of Nov and supplemented with 0 mg/mL, 0.1 mg/mL (35.8% inhibition compared with cell growth in the presence of 75 μg/mL of Nov only at the 15-h time point [*P* < 0.0001 by Student's *t* test]), 0.2 mg/mL (33.1% inhibition compared with cell growth in the presence of 75 μg/mL of Nov only at the 15-h time point [*P* < 0.0001 by Student's *t* test]), 0.5 mg/mL (39.8% inhibition compared with cell growth in the presence of 75 μg/mL of Nov only at the 15-h time point [*P* < 0.0001 by Student's *t* test]), 1.0 mg/mL (45.3% inhibition compared with cell growth in the presence of 75 μg/mL of Nov only at the 15-h time point [*P* < 0.0001 by Student's *t* test]), 1.5 mg/mL (54.3% inhibition compared with cell growth in the presence of 75 μg/mL of Nov only at the 15-h time point [*P* < 0.0001 by Student's *t* test]), or 2.0 mg/mL (100% inhibition compared with cell growth in the presence of 75 μg/mL of Nov only at the 15-h time point [*P* < 0.0001 by Student's *t* test]) of CASP. (C) Growth of cells in the presence of 4 μg/mL of streptomycin (Strep). The MIC of streptomycin for *E. coli* BL21(DE3) Δ*acrB*/pACYCΩ*mtrCDE* is 8 μg/mL. The growth curves of cells supplemented with 0, 0.1, 0.2, 0.5, 1.0, 1.5, and 2.0 mg/mL of CASP are colored as indicated in the key.

We then tested the growth of *Escherichia coli* BL21(DE3) Δ*acrB*/pACYCΩ*mtrCDE* cells, which express the MtrCDE tripartite efflux complex (strain FA19). We used Nov as a test antimicrobial substrate for MtrD as the loss of this protein in *N. gonorrhoeae* due to the genetic inactivation of *mtrD* results in increased susceptibility to this antibiotic (Table S2).

We determined that the levels of growth of *E. coli* BL21(DE3) Δ*acrB*/pACYCΩ*mtrCDE* cells in the presence of 75 μg/mL of Nov (the MIC for Nov was 125 μg/mL) and 2.0 mg/mL of ADP1, ADP2, ADP3, ADP4, or ADP5 were lower by 34.4%, 49.8%, 6.5%, 1.2%, or 47.6%, respectively, than the levels of growth of BL21(DE3) Δ*acrB*/pACYCΩ*mtrCDE* cells in the presence of 75 μg/mL Nov only (Fig. 1A). Interestingly, these *E. coli* cells cannot be grown easily, with approximately 92.1% inhibition, in the presence of 75 μg/mL of Nov and 2.0 mg/mL of the peptide ADP6 (Fig. 1A).

Based on the cell growth study, we chose the peptide ADP6 for further investigation as it was the most potent of the six peptides tested (Fig. 1A). To further enhance the stability of ADP6, we covalently cyclized this peptide by cross-linking its N- and C-terminal cysteines to form CASP (Fig. 1B). CASP was found to be more efficient than ADP6, capable of completely inhibiting the growth of *E. coli* BL21(DE3) Δ*acrB*/pACYCΩ*mtrCDE* cells in the presence of Nov (Fig. 1B), suggesting that it could serve as a potent antibiotic adjuvant. Critically, CASP alone is not toxic to *E. coli* BL21(DE3) Δ*acrB*/pACYCΩ*mtrCDE* cells (Fig. S1). Furthermore, CASP did not render *E. coli* BL21

(DE3) Δ*acrB*/pACYCΩ*mtrCDE* cells more susceptible to an antibiotic (streptomycin) not exported by the gonococcal MtrCDE efflux pump (Fig. 1C). We suggest that since this peptide does not interfere with cell growth, it may provide less selective pressure for the development of bacterial resistance (40, 41).

**Structure of MtrD$_{CR103}$ in complex with the CASP peptide.** To determine how CASP interferes with the function of MtrD, we employed a cryo-EM strategy used previously by us to ascertain how $\beta$-lactams and macrolides interact with amino acids in the deep binding pocket of RND multidrug efflux pumps (13, 42–44). Briefly, we overproduced MtrD$_{CR103}$ in *E. coli* BL21(DE3) Δ*acrB* cells and purified this protein using a Ni$^{2+}$ affinity column. We reconstituted the purified MtrD$_{CR103}$ pump into lipidic nanodiscs. We then incubated 2 $\mu$M the MtrD$_{CR103}$-nanodisc sample with 100 $\mu$M CASP for 2 h to form the MtrD$_{CR103}$-CASP complex and determined the MtrD$_{CR103}$-CASP structure using single-particle cryo-EM (Fig. S2). The reconstituted sample led to a cryo-EM map at a nominal resolution of 2.95 Å (Fig. 2, Fig. S2, and Table S3), allowing us to obtain a structural model of this pump-peptide complex.

The chemical structure of CASP is shown in Fig. 2A. The overall structure of MtrD$_{CR103}$-CASP is very similar to the previously determined MtrD$_{CR103}$ structures in complex with erythromycin or ampicillin (13). In MtrD$_{CR103}$-CASP, the three MtrD$_{CR103}$ protomers display the typical "access," "binding," and "extrusion" conformations (Fig. 2B and C). An extra density corresponding to the CASP molecule is observed to anchor at the distal drug binding pocket of the binding protomer (Fig. 2C to E). No extra densities are found in the access and extrusion protomers. The bound peptide occupies a substantial portion of the internal volume formed by the periplasmic domain of MtrD$_{CR103}$. It appears that a number of residues are engaged in anchoring this peptide, including those within the entrance drug binding site, F-loop, proximal drug binding site, G-loop, distal drug binding site, and hydrophobic patch. Within 4.5 Å of bound CASP, there are 19 residues participating in binding. These residues are H46, F136, I139, M141, R174, F176, S275, T277, A288, M290, Y325, F568, I605, V607, F610, F612, F623, I625, and L668 (Fig. 2E). Particularly, we found that the side chains of R174 and T277 are involved in forming two hydrogen bonds with bound CASP to further secure peptide binding. This observation indeed highlights the importance of residue R174, which was previously found to be critical for substrate recognition (13).

Superimposition of the binding, extrusion, and access protomers of MtrD$_{CR103}$ depicts that there are significant conformational differences between these protomers (Fig. S3), particularly the structural elements making up the periplasmic entrance (formed by the PC1 and PC2 subdomains), where this entrance can be open or closed to help advance the transport cycle.

**Structures of MtrD$_{CR103}$ in complex with other cationic peptides.** The structures of RND multidrug efflux pumps bound with small molecules have been studied extensively (13, 42–48). This structural information has led us to understand the molecular mechanisms employed by these pumps to recognize small-drug molecules and toxic compounds. However, there is little information concerning how these RND pumps recognize and interact with antimicrobial peptides. For example, the *N. gonorrhoeae* MtrD multidrug efflux pump transporter is capable of mediating resistance to host-derived antimicrobial peptides such as cathelicidins and peptide-based antibiotics such as polymyxins (2, 14–17). How this pump specifically interacts with these peptides, however, is not yet understood. To further elucidate the structural basis of pump-peptide interactions and to compare their binding to MtrD-CASP, we solved the cryo-EM structures of the MtrD$_{CR103}$ pump bound with the peptide-based antimicrobial colistin (Col) and the human cathelicidin LL-37 (residues 17 to 32) peptide (LL).

**(i) Structure of MtrD$_{CR103}$-Col.** To study MtrD$_{CR103}$-Col interactions, we incubated 2 $\mu$M the MtrD$_{CR103}$-nanodisc sample with 20 $\mu$M Col for 2 h to form the MtrD$_{CR103}$-Col complex. We obtained a cryo-EM map of this complex at a 3.08-Å resolution (Fig. 3, Fig. S4, and Table S3). The chemical structure of Col is shown in Fig. 3A. Similar to MtrD$_{CR103}$-CASP, the structure of MtrD$_{CR103}$-Col reveals an asymmetric trimer, with the three MtrD$_{CR103}$ protomers exhibiting the access, binding, and extrusion conformational states

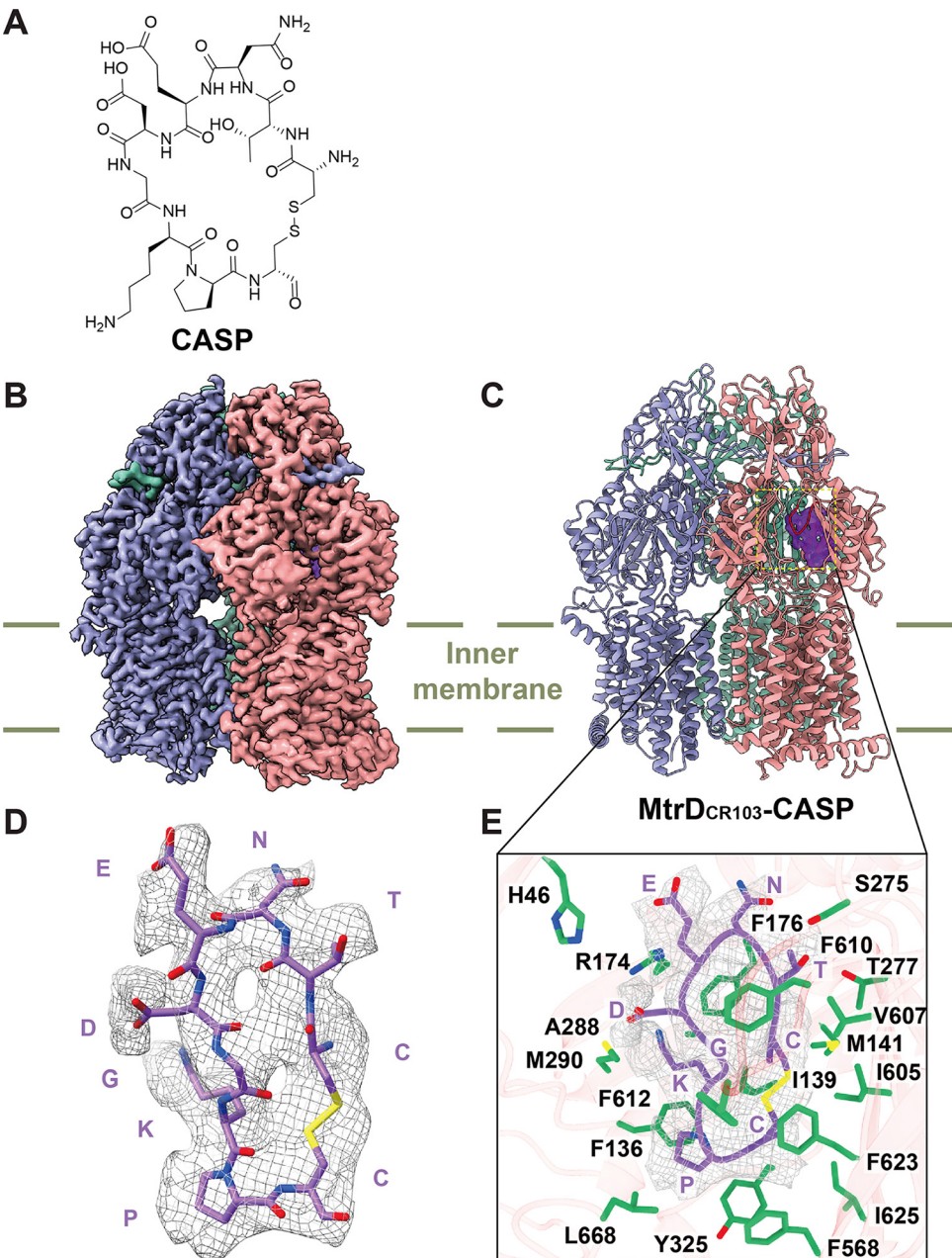

**FIG 2** Cryo-EM structure of the MtrD_{CR103} multidrug efflux pump bound with CASP. (A) Chemical structure of CASP. (B) Side view of the cryo-EM map of the MtrD_{CR103}-CASP complex. The three MtrD_{CR103} protomers are shown in pink (binding protomer), slate (access protomer), and green (extrusion protomer). (C) Ribbon diagram of MtrD_{CR103}-CASP viewed from the membrane plane. The three MtrD_{CR103} protomers are shown in pink (binding protomer), slate (access protomer), and green (extrusion protomer). The cryo-EM density originating from the CASP peptide (purple) is located at the distal drug binding site. The G-loop of the binding protomer is shown in red. (D) Cryo-EM density of bound CASP. CASP is shown as purple sticks. The density of CASP is shown as gray mesh. (E) Enlarged view of the CASP binding site. Residues participating in CASP binding are shown as green sticks. The cryo-EM density of bound CASP is shown as gray mesh. The bound CASP peptide is represented by purple sticks.

(Fig. 3B and C). In the binding protomer of MtrD_{CR103}, the cryo-EM map depicts an additional large density corresponding to the bound Col peptide (Fig. 3C to E). This bound Col molecule is found deep inside the distal drug binding site of the pump (Fig. 3E). The Col binding site overlaps that of CASP; however, it appears that MtrD_{CR103} utilizes a slightly different subset of residues to attach this peptide. Again, the nature of this protein-peptide interaction is mostly hydrophobic. Within 4.5 Å of this bound peptide, there

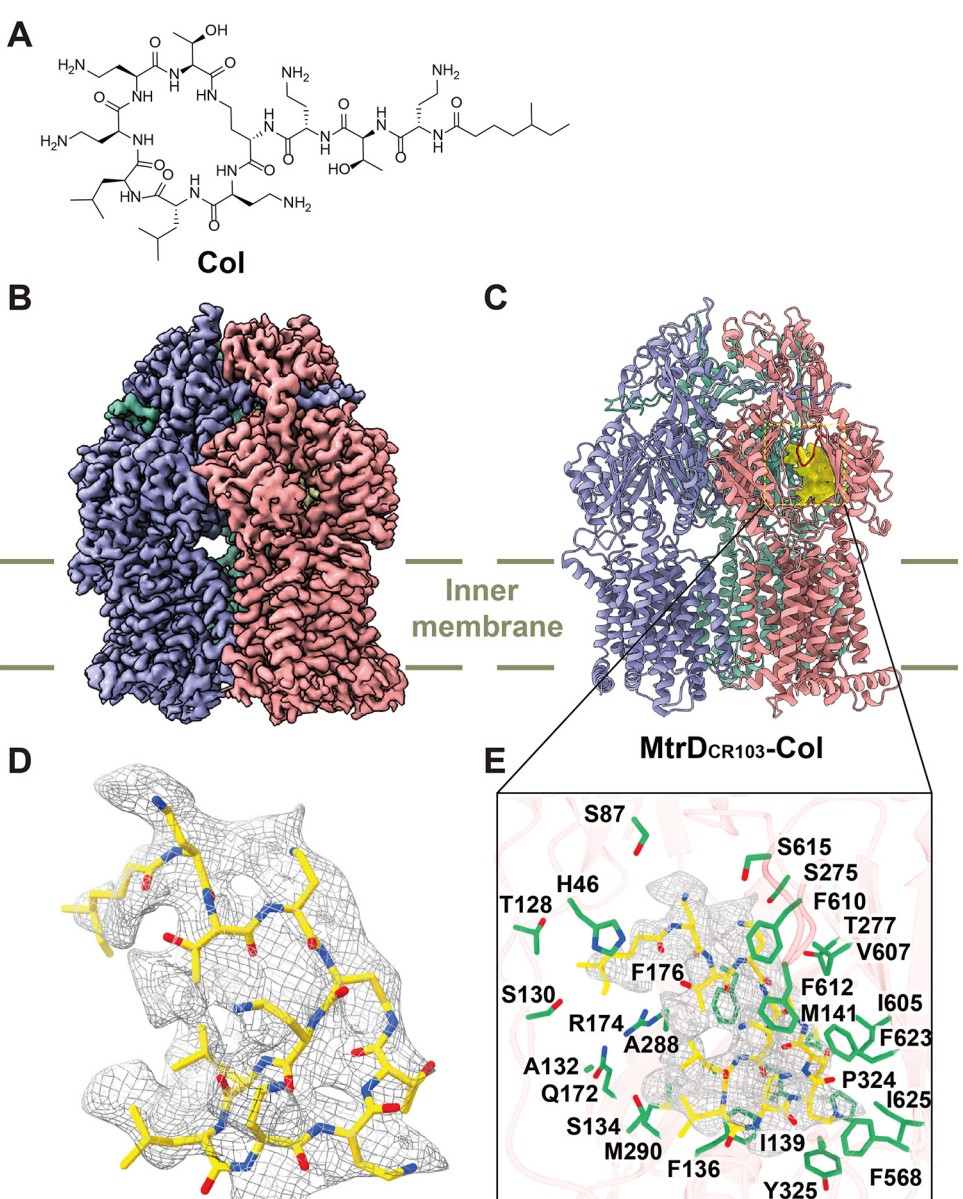

**FIG 3** Cryo-EM structure of the MtrD_{CR103} multidrug efflux pump bound with the colistin (Col) peptide. (A) Chemical structure of Col. (B) Side view of the cryo-EM map of the MtrD_{CR103}-Col complex. The three MtrD_{CR103} protomers are shown in pink (binding protomer), slate (access protomer), and green (extrusion protomer). (C) Ribbon diagram of MtrD_{CR103}-Col viewed from the membrane plane. The three MtrD_{CR103} protomers are shown in pink (binding protomer), slate (access protomer), and green (extrusion protomer). The cryo-EM density originating from the Col antimicrobial peptide (yellow) is located at the distal drug binding site. The G-loop of the binding protomer is shown in red. (D) Cryo-EM density of bound Col. Col is shown as yellow sticks. The density of Col is shown as gray mesh. (E) Enlarged view of the Col binding site. Residues participating in Col binding are shown as green sticks. The cryo-EM density of bound Col is shown as gray mesh. The bound Col peptide is represented by yellow sticks.

are 26 residues, H46, S87, T128, S130, A132, S134, F136, I139, M141, Q172, R174, F176, S275, T277, A288, M290, P324, Y325, F568, I605, V607, F610, F612, S615, F623, and I625, that participate in facilitating Col binding (Fig. 3E). Many of these residues are in common with those for CASP binding. One notable difference is that some of the polar residues, such as S87, T128, S130, S134, Q172, and S615, do not interact with CASP. Presumably, these residues are more specific for Col recognition.

**(ii) Structure of MtrD_{CR103}-LL.** As the MtrD multidrug efflux pump recognizes and confers resistance to the human antimicrobial peptide LL-37 (14), we decided to define

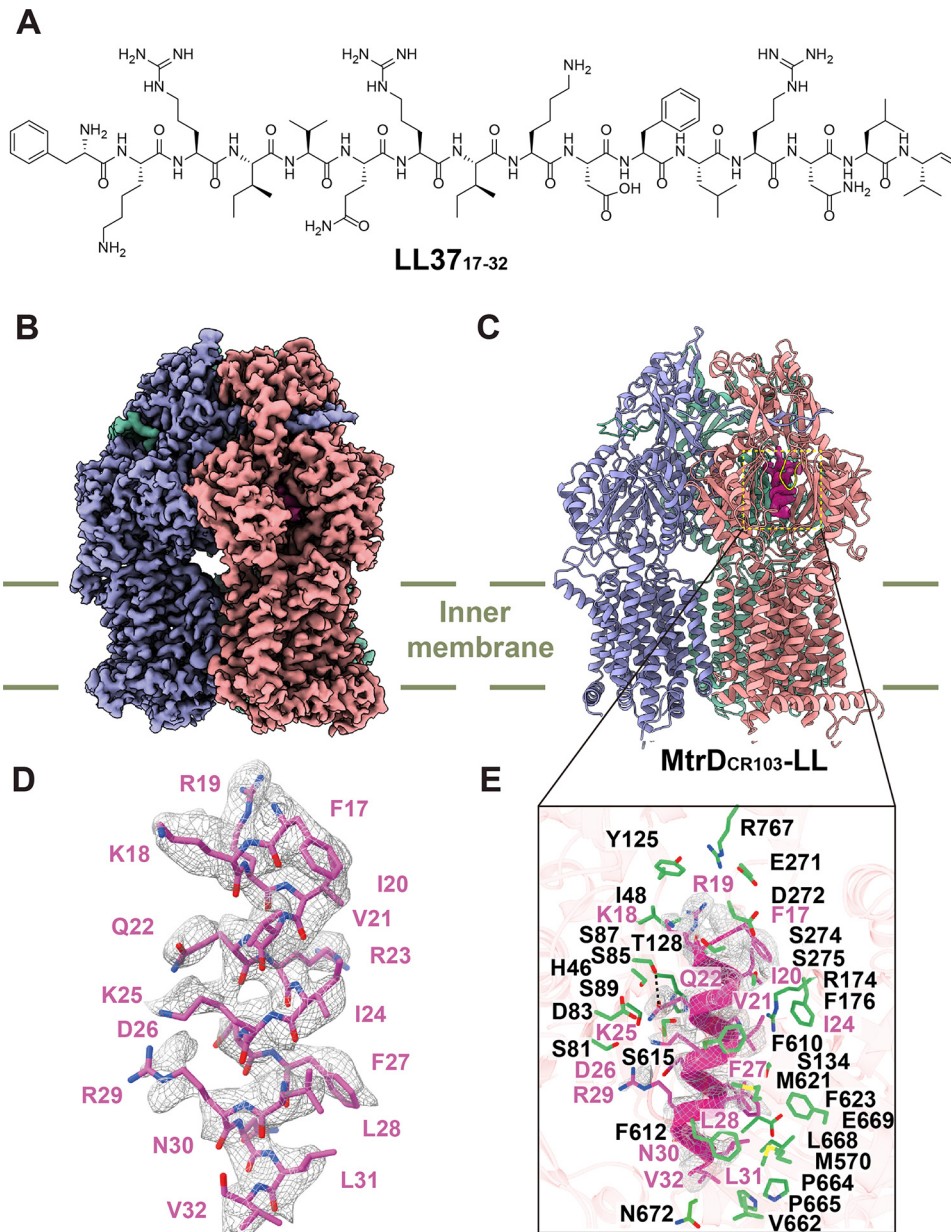

**FIG 4** Cryo-EM structure of the MtrD_{CR103} multidrug efflux pump bound with the human cathelicidin LL-37 (residues 17 to 32) peptide (LL). (A) Chemical structure of LL. (B) Side view of the cryo-EM map of the MtrD_{CR103}-LL complex. The three MtrD_{CR103} protomers are shown in pink (binding protomer), slate (access protomer), and green (extrusion protomer). (C) Ribbon diagram of MtrD_{CR103}-LL viewed from the membrane plane. The three MtrD_{CR103} protomers are shown in pink (binding protomer), slate (access protomer), and green (extrusion protomer). The cryo-EM density originating from the LL cathelicidin peptide (magenta) is located at the distal drug binding site. The G-loop of the binding protomer is shown in yellow. (D) Cryo-EM density of bound LL. LL is shown as magenta sticks. The density of LL is shown as gray mesh. (E) Enlarged view of the LL binding site. Residues participating in LL binding are shown as green sticks. The cryo-EM density of bound Col is shown as gray mesh. The bound LL peptide is represented by a magenta helix.

the cryo-EM structure of MtrD_{CR103} bound by LL-37. We synthesized an all-$\alpha$-helical peptide (LL) that contains 16 amino acids of LL-37 (residues 17 to 32) (16). This portion of LL-37 has been shown to be sufficient to kill *N. gonorrhoeae* FA19 cells (49). Similar to the procedure used to create the MtrD_{CR103}-CASP and MtrD_{CR103}-Col complexes, we incubated 2 $\mu$M the MtrD_{CR103}-nanodisc sample with 50 $\mu$M LL for 2 h to form the MtrD_{CR103}-LL complex. We then obtained a cryo-EM map of this complex at a 2.89-Å resolution (Fig. 4, Fig. S5, and Table S3).

The chemical structure of LL is depicted in Fig. 4A. The MtrD$_{CR103}$-LL structure also presents an asymmetric trimer, with the three protomers displaying the distinctive binding, extrusion, and access conformations (Fig. 4B and C). Superimposition of the MtrD$_{CR103}$-LL trimer with those of MtrD$_{CR103}$-CASP and MtrD$_{CR103}$-Col gives rise to root mean standard deviation (RMSD) values within 0.6 Å, suggesting that the conformations of these three trimers are more or less identical. Again, within the binding protomer of MtrD$_{CR103}$, the cryo-EM images depict an additional large density corresponding to the bound α-helical LL peptide located inside the open cleft of the periplasmic domain (Fig. 4C to E). The LL binding site is very spacious. Our cryo-EM structure indicates that the bound LL peptide occupies spaces that form the entrance, proximal, and distal drug binding sites (Fig. 4E), where this α-helical peptide spans both the binding and extrusion tunnels of the MtrD$_{CR103}$ pump. The binding of LL is expansive. Within 4.5 Å of the bound peptide, there are 31 residues that secure the binding. These residues include H46, I48, S81, D83, S85, S87, S89, S116, Y125, T128, S130, S134, R174, F176, E271, D272, S274, S275, M570, F610, F612, S615, M621, F623, V662, P664, P665, L668, E669, N672, and R767 (Fig. 4E). In particular, D83 forms a salt bridge and S87 form a hydrogen bond with the LL peptide. Of the 31 binding residues, only 7 residues (H46, R174, F176, S275, F610, F612, and F623) are also involved in Col and CASP binding. Superimposition of the binding sites of LL, Col, and CASP indicates that MtrD$_{CR103}$ binds Col and CASP in a very similar manner (Fig. S6). The majority of these two cyclic peptides are located at the distal drug binding pocket and immediately behind the G-loop. However, the binding mode of the elongated, α-helical LL is quite distinct from those of Col and CASP. LL is found to span both the distal and proximal drug binding sites as well as part of the entrance site. The differences in these binding modes may be attributed in part to the fact that both Col and CASP are cyclic in form, but LL is an all-α-helical peptide.

**CASP action against *N. gonorrhoeae* and other Gram-negative bacteria.** We tested whether incubation of *N. gonorrhoeae* strain FA1090 with CASP would render gonococci susceptible to sublethal levels of Nov (0.06 and 0.125 μg/mL). We found that CASP at 2.0 mg/mL in GCB (gonococcal base) broth did not impede the growth of FA1090. However, CASP also did not sensitize FA1090 to the lethal effects of Nov, although Nov is a substrate for the MtrCDE efflux pump (Fig. S7). Similarly, CASP could not sensitize two different gonococcal strains (FA19 and CDC2) to Nov (data not shown). Based on these results, we concluded that while CASP could sensitize *E. coli* cells producing the gonococcal MtrCDE efflux pump, it did not have a similar effect on *N. gonorrhoeae*.

Furthermore, we also found that CASP could not sensitize a nonpathogenic commensal *Neisseria* sp. (*N. subflava* [ATCC 49275]) to Nov (Fig. S8). As *E. coli* is a Gram-negative bacillus, but *N. gonorrhoeae* and *N. subflava* are Gram-negative cocci, we hypothesized that these heptapeptides may be more specific for Gram-negative bacilli. Since *N. gonorrhoeae* MtrD is homologous to the *E. coli* AcrB, *Pseudomonas aeruginosa* MexB, and *Salmonella enterica* serovar Typhimurium AcrB multidrug efflux pumps, we rationalized that this cyclic peptide could be capable of inhibiting the growth of these Gram-negative bacilli in the presence of Nov.

To elucidate if the CASP peptide is capable of binding other RND multidrug efflux pumps such as *E. coli* AcrB and *P. aeruginosa* MexB, we used the program AutoDock Vina (50) to study AcrB-CASP and MexB-CASP interactions. We also docked this CASP peptide into the *N. gonorrhoeae* MtrD$_{CR103}$ multidrug binding site. We found that all of these multidrug efflux pumps are able to specifically contact this cyclic peptide, with predicted binding affinities of between −6.6 and −9.4 kcal/mol (Fig. S9).

To further ensure that these RND pumps are targets of CASP, we utilized *E. coli* K-12 Δ*tolC::kan* cells (strain JW5503), which have a deletion of the *tolC* gene expressing the TolC outer membrane channel. *E. coli* harbors seven RND membrane proteins of the hydrophobe-amphiphile efflux (HAE) (AcrB, AcrD, AcrF, MdtB, MdtC, and YhiV) and heavy metal efflux (HME) (CusA) types (51–59). All of these RND proteins could be

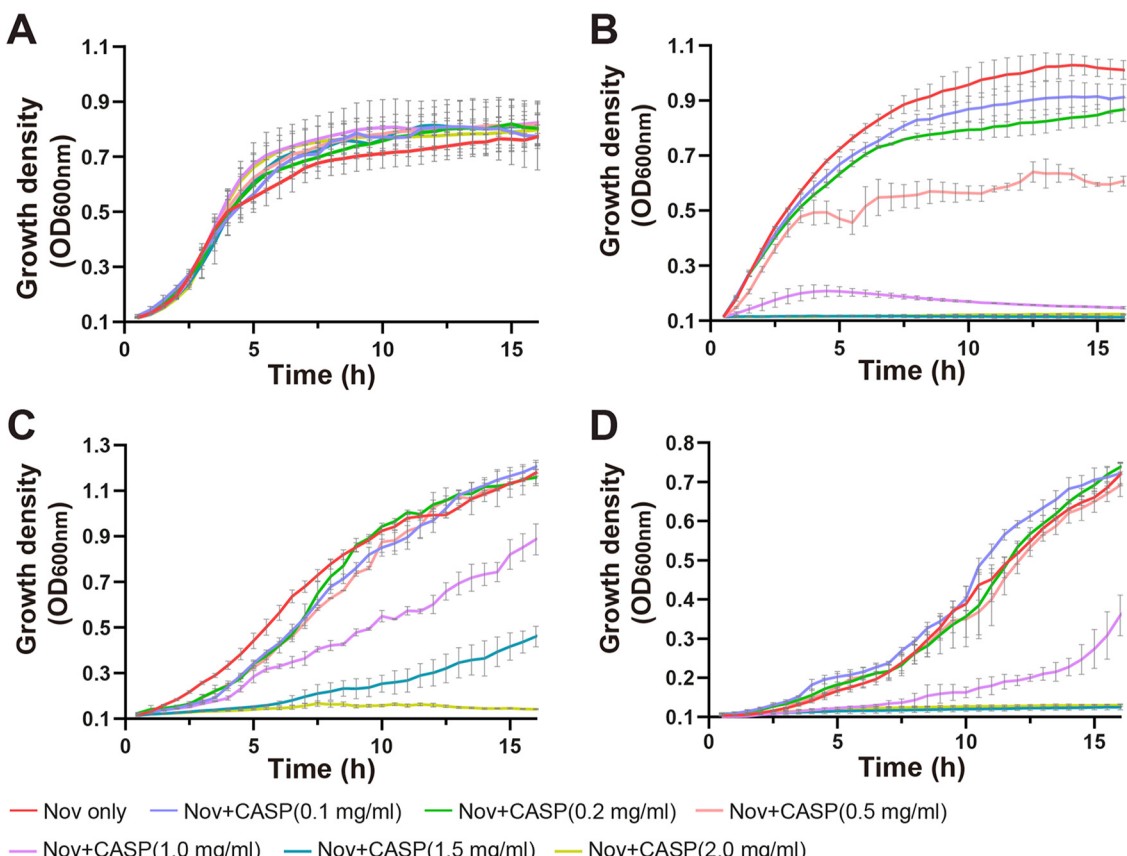

Legend:
— Nov only   — Nov+CASP(0.1 mg/ml)   — Nov+CASP(0.2 mg/ml)   — Nov+CASP(0.5 mg/ml)
— Nov+CASP(1.0 mg/ml)   — Nov+CASP(1.5 mg/ml)   — Nov+CASP(2.0 mg/ml)

**FIG 5** Growth of Gram-negative bacilli in the presence of Nov with or without CASP. (A) Growth of *E. coli* K-12 Δ*tolC::kan* cells in the presence of 2 μg/mL of Nov. The MIC of Nov for *E. coli* K-12 Δ*tolC::kan* is 4 μg/mL in this study. The growth curves of cells supplemented with 0, 0.1, 0.2, 0.5, 1.0, 1.5, and 2 mg/mL of CASP are colored as indicated in the key at the bottom. (B) Growth of *E. coli* K-12 (BW25113) cells in the presence of 75 μg/mL of Nov. Cells were grown in LB supplemented with 0 mg/mL, 0.1 mg/mL (11.5% inhibition compared with cell growth in the presence of 75 μg/mL of Nov only at the 15-h time point [*P* = 0.0005 by Student's *t* test]), 0.2 mg/mL (19.0% inhibition compared with cell growth in the presence of 75 μg/mL of Nov only at the 15-h time point [*P* < 0.0001 by Student's *t* test]), 0.5 mg/mL (46.8% inhibition compared with cell growth in the presence of 75 μg/mL of Nov only at the 15-h time point [*P* < 0.0001 by Student's *t* test]), 1.0 mg/mL (96.1% inhibition compared with cell growth in the presence of 75 μg/mL of Nov only at the 15-h time point [*P* < 0.0001 by Student's *t* test]), 1.5 mg/mL (100% inhibition compared with cell growth in the presence of 75 μg/mL of Nov only at the 15-h time point [*P* < 0.0001 by Student's *t* test]), or 2.0 mg/mL (100% inhibition compared with cell growth in the presence of 75 μg/mL of Nov only at the 15-h time point [*P* < 0.0001 by Student's *t* test]) of CASP. (C) Growth of *Salmonella enterica* subsp. *enterica* (ATCC 53648) cells in the presence of 128 μg/mL of Nov. Cells were grown in LB supplemented with 0 mg/mL, 0.1 mg/mL (0% inhibition compared with cell growth in the presence of 128 μg/mL of Nov only at the 15-h time point [*P* = 0.9291 by Student's *t* test]), 0.2 mg/mL (1.3% inhibition compared with cell growth in the presence of 128 μg/mL of Nov only at the 15-h time point [*P* = 0.9978 by Student's *t* test]), 0.5 mg/mL (1.0% inhibition compared with cell growth in the presence of 128 μg/mL of Nov only at the 15-h time point [*P* = 0.9996 by Student's *t* test]), 1.0 mg/mL (31.6% inhibition compared with cell growth in the presence of 128 μg/mL of Nov only at the 15-h time point [*P* < 0.0001 by Student's *t* test]), 1.5 mg/mL (70.3% inhibition compared with cell growth in the presence of 128 μg/mL of Nov only at the 15-h time point [*P* < 0.0001 by Student's *t* test]), or 2.0 mg/mL (97.2% inhibition compared with cell growth in the presence of 128 μg/mL of Nov only at the 15-h time point [*P* < 0.0001 by Student's *t* test]) of CASP. (D) Growth of *Pseudomonas aeruginosa* (ATCC 39327) cells in the presence of 512 μg/mL of Nov. Cells were grown in LB supplemented with 0 mg/mL, 0.1 mg/mL (0% inhibition compared with cell growth in the presence of 512 μg/mL of Nov only at the 15-h time point [*P* = 0.1617 by Student's *t* test]), 0.2 mg/mL (0% inhibition compared with cell growth in the presence of 512 μg/mL of Nov only at the 15-h time point [*P* = 0.4612 by Student's *t* test]), 0.5 mg/mL (1.8% inhibition compared with cell growth in the presence of 512 μg/mL of Nov only at the 15-h time point [*P* = 0.9940 by Student's *t* test]), 1.0 mg/mL (69.9% inhibition compared with cell growth in the presence of 512 μg/mL of Nov only at the 15-h time point [*P* < 0.0001 by Student's *t* test]), 1.5 mg/mL (96.9% inhibition compared with cell growth in the presence of 512 μg/mL of Nov only at the 15-h time point [*P* < 0.0001 by Student's *t* test]), or 2.0 mg/mL (95.9% inhibition compared with cell growth in the presence of 512 μg/mL of Nov only at the 15-h time point [*P* < 0.0001 by Student's *t* test]) of CASP.

subjected to interactions with the CASP peptide. It appears that all six HAE-RND pumps are coordinated with TolC to function. We therefore grew *E. coli* K-12 Δ*tolC::kan* cells in the presence of 2 μg/mL Nov (the MIC of Nov for *E. coli* K-12 Δ*tolC::kan* was found to be 4 μg/mL in this study), supplemented with different concentrations of CASP (0, 0.1, 0.2, 0.5, 1.0, 1.5, and 2.0 mg/mL). We found that CASP could not sensitize *E. coli* K-12 Δ*tolC::kan* cells to Nov (Fig. 5A).

After confirming that RND multidrug efflux pumps are targets of the CASP peptide, we then turned to strains of *E. coli* K-12 (BW25113), *Salmonella enterica* subsp. *enterica* (ATCC 53648), and *Pseudomonas aeruginosa* (ATCC 39327). We grew *E. coli* K-12 cells in the presence of 75 $\mu$g/mL Nov (the MIC of Nov for *E. coli* K-12 was found to be >128 $\mu$g/mL in this study), supplemented with 0, 0.1, 0.2, 0.5, 1.0, 1.5, or 2.0 mg/mL CASP (Fig. 5B). We observed that *E. coli* K-12 cells have a severe growth defect in the presence of 75 $\mu$g/mL of Nov and 1.0, 1.5, or 2.0 mg/mL of the CASP peptide, suggesting that this cyclic peptide can act as an antimicrobial adjuvant to effectively inhibit the growth of *E. coli* K-12 cells. We also tested the growth of *S. enterica* subsp. *enterica* cells in the presence of 128 $\mu$g/mL Nov (the MIC of Nov for *S. enterica* subsp. *enterica* was found to be 512 $\mu$g/mL in this study), supplemented with 0, 0.1, 0.2, 0.5, 1.0, 1.5, or 2.0 mg/mL CASP (Fig. 5C). We found that the growth of *S. enterica* subsp. *enterica* cells is inhibited in the presence of 128 $\mu$g/mL of Nov and 1.5 or 2.0 mg/mL of CASP. In addition, we grew *P. aeruginosa* cells in the presence of 512 $\mu$g/mL Nov (the MIC of Nov for *P. aeruginosa* was found to be >1,000 $\mu$g/mL in this study), supplemented with 0, 0.1, 0.2, 0.5, 1.0, 1.5, or 2.0 mg/mL CASP (Fig. 5D). Similarly, we found that *P. aeruginosa* cells cannot be grown in the presence of 512 $\mu$g/mL of Nov and 1.0, 1.5, or 2.0 mg/mL of the CASP peptide.

## DISCUSSION

Emerging multidrug resistance is threatening our ability to treat bacterial infections. Pathogens such as *N. gonorrhoeae* and many other Gram-negative pathogens are expected to become increasingly resistant and possibly untreatable within the next decade. According to the CDC's antibiotic resistance threat report (60), there are more than 2.8 million antibiotic-resistant infections that occur in the United States each year, and over 35,000 cases are fatal as a result of these diseases. One of the most important resistance mechanisms that Gram-negative bacterial pathogens use to counteract the action of antimicrobials is the action of multidrug efflux pumps, which efficiently diminish intracellular drug concentrations to the level where these pathogens can easily evade their antibacterial activity.

We previously solved cryo-EM structures of MtrD$_{CR103}$, carrying a mosaic-like sequence, in complex with erythromycin or hydrolyzed ampicillin (13). It was observed that both drugs are bound at the distal pocket of MtrD$_{CR103}$. Their binding sites overlap each other. However, it appears that MtrD$_{CR103}$ utilizes similar but slightly distinct subsets of residues to anchor these drugs. Based on the cryo-EM structures of MtrD$_{CR103}$-CASP, MtrD$_{CR103}$-Col, and MtrD$_{CR103}$-LL, we observed that MtrD$_{CR103}$ employs different binding modes to anchor linear and cyclic peptides. The binding of the linear human cathelicidin LL peptide is accomplished by a collaborative effort among residues creating the entrance, proximal, distal, and hydrophobic patched drug binding sites. This binding is expansive and involves at least 31 residues spanning different drug binding sites. However, for cyclic peptide binding, the bound cyclic peptide CASP and the polycationic peptide Col are largely localized at the distal drug binding pocket. This binding mode is more familiar and similar to that of canonical antibiotic binding for these types of pumps. Although the binding modes for the linear and cyclic antimicrobial peptides are different, MtrD$_{CR103}$ does utilize several common residues to secure the binding of these peptides. These are the positively charged residues H46 and R174, the polar residue S275, and the aromatic residues F176, F610, F612, and F623. Interestingly, mutations on residues R174, F176, F610, F612, and F623 have been shown to have a direct impact on antimicrobial resistance (AMR) (13, 17). In addition, F176 and F610 contribute to the critical distal hydrophobic patch that strongly influences drug recognition (13). Therefore, these residues could be important for recognizing multiple agents, including both antibiotics and antimicrobial peptides.

Presently, we cannot explain why CASP treatment of Gram-negative bacilli can sensitize a panel of bacteria to Nov but fails to do so against *N. gonorrhoeae* or *N. subflava*. We hypothesize, however, that differences in the permeability barrier of the outer membranes of rod-shaped bacilli versus coccoid-shaped *Neisseria* may, in part, determine the

ability of CASP to enter and reach MtrD-like transporter proteins harbored by these different bacteria. Despite this, we hope that this study may help initiate structure-guided and peptide-based antimicrobial design for obstructing these critical multidrug RND (and other) efflux pumps that help mediate AMR, allowing the return of previously used antibiotics. Thus, CASP-like compounds could serve as an adjunctive therapy to combat the increasing threat of multidrug-resistant (MDR) pathogens.

## MATERIALS AND METHODS

**Expression and purification of MtrD.** The *mtrD* gene, encoding the MtrD multidrug efflux pump, from *N. gonorrhoeae* strain FA19 was cloned into the pET15b expression vector in frame with a 6×His tag at the C terminus. The resulting pET15bΩ*mtrD* plasmid was confirmed by the Sanger method of DNA sequencing. The MtrD protein was overproduced in *E. coli* BL21(DE3) Δ*acrB* cells, which harbor a deletion in the chromosomal *acrB* gene. Cells were grown in 6 L of Luria broth (LB) medium with 100 $\mu$g/mL ampicillin at 37°C. When the optical density at 600 nm (OD$_{600}$) reached 0.5, the culture was treated with 0.2 mM isopropyl-$\beta$-D-thiogalactopyranoside (IPTG) to induce *mtrD*$_{CR103}$ expression. The cells were then harvested within 3 h of induction. The collected bacteria were resuspended in low-salt buffer (100 mM sodium phosphate [pH 7.4], 10% glycerol, 5 mM EDTA, and 1 mM phenylmethanesulfonyl fluoride [PMSF]) and then disrupted with a French pressure cell. The membrane fraction was collected and washed twice with high-salt buffer (20 mM sodium phosphate [pH 7.4], 2 M KCl, 10% glycerol, 5 mM EDTA, and 1 mM PMSF) and once with 20 mM HEPES-NaOH buffer (pH 7.5) containing 20 mM NaCl and 1 mM PMSF. The membrane protein was solubilized in 2% (wt/vol) *n*-dodecyl-$\beta$-D-maltoside (DDM), and the insoluble material was subsequently removed by ultracentrifugation at 100,000 × *g*. The extracted protein was then purified with a Ni$^{2+}$ affinity column. The purity of the MtrD$_{CR103}$ protein (>95%) was judged using an SDS-PAGE gel stained with Coomassie brilliant blue. The purified protein was dialyzed against 20 mM Na-HEPES (pH 7.5) and concentrated to 5 mg/mL in a buffer containing 100 mM NaHCO$_3$ (pH 8.6) and 0.05% DDM.

**Screening of heptapeptides that specifically bind MtrD.** The PhD-C7C phage display peptide library (New England BioLabs [NEB]) was used for screening antimicrobial peptides specific for the MtrD pump (strain FA19). The randomized segment of the PhD-C7C library is flanked by a pair of cysteine residues that are oxidized during phage assembly to a disulfide linkage, resulting in the displayed peptides being presented to the target as cyclized peptides. Peptide-harboring phage selection was performed as described below. A 96-well plate was coated with 1 mg/mL of the purified MtrD protein in a buffer containing 100 mM NaHCO$_3$ (pH 8.6) and 0.05% DDM and incubated overnight at 4°C. The coating solution was poured off, and blocking buffer (0.1 M NaHCO$_3$ [pH 8.6] and 5 mg/mL bovine serum albumin [BSA]) was added for 1 h at 4°C. After washing 6 times with TBST (Tris-buffered saline [TBS] plus 0.1% [vol/vol] Tween 20), 100 $\mu$L (2 × 10$^9$ PFU) of the library was added to the plate well for 2 h at room temperature, with gentle agitation. The plate well was washed 10 times with TBST to remove unbound phages. The bound phages were eluted with 0.2 M glycine-HCl (pH 2.2) and neutralized with 1 M Tris-HCl (pH 9.1). Eluted phages were amplified in 20 mL LB inoculated with the *E. coli* ER2738 strain, purified by polyethylene glycol (PEG)-NaCl precipitation, and titrated according to the NEB standard protocol to be used for the next round of selection. The two next rounds of selection were performed under more stringent conditions as the concentration of Tween 20 was increased (0.5%). After the third round of selection, eluted phages were plated onto NZ-amide and yeast extract (NZY) plates containing 20 $\mu$g/mL tetracycline. Individual randomly selected bacterial colonies were cultured and used for the isolation of phage DNA according to standard phage DNA purification procedures. The −96 gIII sequencing primer (CCCTCATAGTTAGCGTAACG) was used for sequencing to determine the amino acid sequences of inserts in the selected phage.

The third-round-screened peptides listed in Table S1 in the supplemental material were subsequently synthesized by solid-phase peptide synthesis (GenScript). As a first screen to assess their activity, we studied the growth curve of *E. coli* BL21(DE3) Δ*acrB*/pACYCΩ*mtrCDE* cells, as a surrogate for the *N. gonorrhoeae* strain FA19 *mtrCDE* genes, in the presence of each peptide with 75 $\mu$g/mL novobiocin. Of the peptides, only peptide 6 (ADP6) showed significant inhibition of the MtrCDE efflux system. To improve the stability of ADP6, the head-to-tail cyclic peptide ADP6 was synthesized (GenScript USA, Piscataway, NJ) to produce the CASP peptide.

**Determination of Nov MICs against gonococci.** The MICs for Nov were determined by agar dilution as described previously (53). The construction of the *mtrD::kan* transformants was described previously (3, 61); note that WHO X was originally termed H041.

**Growth and CASP treatment of *E. coli* BL21(DE3) Δ*acrB*/pACYCΩ*mtrCDE* cells.** The construction and expression of the tripartite efflux pump system MtrCDE (strain FA19) were described previously by Janganan et al. (62). Briefly, the *mtrC* and *mtrE* genes were cloned into multiple-cloning site 1 (MCS1) of pACYCDuet, whereas *mtrD* was cloned into MCS2 of pACYCDuet, to generate the pACYCΩ*mtrCDE* expression vector. Each expressed protein has an in-frame 6×His tag at the C terminus. Next, *E. coli* BL21 (DE3) Δ*acrB* cells harboring the plasmid pACYCΩ*mtrCDE* were grown in LB medium containing 12.5 $\mu$g/mL chloramphenicol at 37°C to an OD$_{600}$ of 0.4 and induced with 0.2 mM IPTG for 1 h at 37°C. Cells were collected and washed with LB three times to remove chloramphenicol. The cells were then resuspended using LB supplemented with 75 $\mu$g/mL Nov. The OD$_{600}$ was adjusted to 0.1. One hundred microliters of the cell culture was aliquoted into each well of a 96-well plate. Two hundred fifty micrograms of different peptides screened from the phage display library was added to each well containing 100 $\mu$L of the cell culture. The

experiments were performed at 37°C on a microplate reader (BioTek) by monitoring the $OD_{600}$ value, which was recorded every 30 min for 24 h. Each experiment was repeated at least 3 times in duplicate.

For cell growth in the presence of the antibiotic streptomycin, *E. coli* BL21(DE3) $\Delta acrB$/pACYC$\Omega mtrCDE$ cells were grown in LB medium at 37°C to an $OD_{600}$ of 0.6. Cells were collected, washed with LB once, resuspend with LB to an $OD_{600}$ of 1.0, and then further diluted to an $OD_{600}$ of 0.1 with LB supplemented with 4 $\mu$g/mL streptomycin. Cells were inoculated into a 96-well plate with a total volume of 100 $\mu$L. A final concentration of 0, 0.1, 0.2, 0.5, 1.0, 1.5, or 2.0 mg/mL of the CASP peptide was added to each well. The experiments were performed at 37°C on a microplate reader (BioTek). The $OD_{600}$ value was monitored and recorded every 30 min within a 24-h period. Each experiment was repeated a minimum of three times.

**Expression and purification of MtrD$_{CR103}$.** The $mtrD_{CR103}$ gene, encoding the MtrD$_{CR103}$ multidrug efflux pump, from *N. gonorrhoeae* strain CR.103 was cloned into the pET15b expression vector in frame with a 6×His tag at the C terminus. The resulting pET15b$\Omega mtrD_{CR103}$ plasmid was confirmed by the Sanger method of DNA sequencing. The procedures for expressing and purifying the MtrD$_{CR103}$ protein were described previously by Lyu et al. (13). The purity of the MtrD$_{CR103}$ protein (>95%) was judged using an SDS-PAGE gel stained with Coomassie brilliant blue. The purified protein was dialyzed against 20 mM Na-HEPES (pH 7.5) and concentrated to 8.5 mg/mL (78 $\mu$M) in a buffer containing 20 mM Na-HEPES (pH 7.5) and 0.05% DDM.

**Nanodisc preparation.** To assemble MtrD$_{CR103}$ into nanodiscs, a mixture containing 20 $\mu$M MtrD$_{CR103}$, 45 $\mu$M membrane scaffold protein (MSP) (1E3D1), and 930 $\mu$M *E. coli* total lipid extract was incubated for 15 min at room temperature. Afterwards, 0.8 mg/mL of prewashed Bio-beads (Bio-Rad) was added. The resultant mixture was incubated for 1 h on ice, followed by incubation overnight at 4°C. The protein-nanodisc solution was filtered through a 0.22-$\mu$m nitrocellulose filter tube to remove the Bio-beads. To separate free nanodiscs from MtrD$_{CR103}$-loaded nanodiscs, the filtered protein-nanodisc solution was loaded into a Superose 6 column (GE Healthcare) equilibrated with buffer containing 20 mM Tris-HCl (pH 7.5) and 100 mM NaCl. Fractions corresponding to the size of the trimeric MtrD$_{CR103}$-nanodisc complex were collected for cryo-EM studies.

**Electron microscopy sample preparation.** A 2 $\mu$M MtrD$_{CR103}$-nanodisc sample was incubated with 100 $\mu$M CASP, 20 $\mu$M Col, or 50 $\mu$M LL for 2 h to form the MtrD$_{CR103}$-peptide complex. The samples were applied to glow-discharged holey carbon grids (Quantifoil Cu R1.2/1.3, 300 mesh), blotted for 17 s, and then plunge-frozen in liquid ethane using a Vitrobot instrument (Thermo Fisher). The grids were then transferred to cartridges.

For the MtrD$_{CR103}$-CASP complex data set, the images were recorded at a $-0.75$- to $-1.75$-$\mu$m defocus on a K3 summit direct electron detector (Gatan) with counting mode at a nominal magnification of ×81,000, corresponding to a sampling interval of 1.08 Å/pixel (superresolution, 0.54 Å/pixel). Each micrograph was exposed for 2.6 s at a dose rate of 18.0 $e^-$/s/physical pixel (total specimen dose, 40 $e^-$/Å$^2$), and 40 frames were captured per specimen area using Latitude.

For the MtrD$_{CR103}$-Col complex data set, the images were recorded at a $-1.75$- to $-2.5$-$\mu$m defocus on a K3 summit direct electron detector (Gatan) with counting mode at a nominal magnification of ×81,000, corresponding to a sampling interval of 1.12 Å/pixel (superresolution, 0.56 Å/pixel). Each micrograph was exposed for 2.8 s at a dose rate of 17.98 $e^-$/s/physical pixel (total specimen dose, 40 $e^-$/Å$^2$), and 40 frames were captured per specimen area using Latitude.

For the MtrD$_{CR103}$-LL complex data set, the images were recorded at a $-1.0$- to $-2.25$-$\mu$m defocus on a K3 summit direct electron detector (Gatan) with counting mode at a nominal magnification of ×81,000, corresponding to a sampling interval of 1.08 Å/pixel (superresolution, 0.54 Å/pixel). Each micrograph was exposed for 2.6 s at a dose rate of 18.035 $e^-$/s/physical pixel (total specimen dose, 40 $e^-$/Å$^2$), and 40 frames were captured per specimen area using Latitude.

**Data collection and processing.** For MtrD$_{CR103}$-CASP, the image stacks in the superresolution model were aligned using cryoSPARC (63). The contrast transfer function (CTF) parameters of the micrographs were determined using Patch CTF (63). After manual inspection and sorting to discard poor images, ~2,000 particles were manually picked to generate templates for automatic picking. Initially, 2,722,287 particles were selected after autopicking in cryoSPARC (63). Several iterative rounds of two-dimensional (2D) classifications were carried out to remove false picks and classes with unclear features, ice contamination, or carbon. The resulting 586,597 particles were used to generate a reference-free *ab initio* three-dimensional (3D) reconstruction. Two rounds of heterogeneous refinement were used, where 269,948 particles were used for further processing with nonuniform refinement using cryoSPARC (63). The particles were then processed by focused 3D variability using a soft mask covering the binding protomer, where 129,244 particles were chosen for further processing with local CTF refinement and local refinement using cryoSPARC (63), resulting in a 2.95-Å-global-resolution map based on the gold standard Fourier shell correlation (FSC) (Table S3 and Fig. S2).

For MtrD$_{CR103}$-Col, 7,449,604 particles were selected after autopicking in cryoSPARC (63). After several rounds of 2D classifications, 1,060,087 particles were selected to generate three *ab initio* models and then subjected to two rounds of 3D heterogeneous refinements. The resulting 584,861 particles were finally chosen for nonuniform refinement. The particles were then processed by focused 3D variability using a soft mask covering the binding protomer, where 301,061 particles were chosen for further processing with local CTF refinement and local refinement using cryoSPARC (63), resulting in a 3.08-Å-resolution cryo-EM structure (Table S3 and Fig. S4).

For MtrD$_{CR103}$-LL, the Topaz pipeline program was used for initial particle picking. Approximately 100 micrographs were chosen to create the Topaz training model. A total of 15,024,699 particles were selected using Topaz extract in cryoSPARC (63). After several rounds of 2D classification and the process

of removing duplicate particles, 1,630,817 particles were selected to generate three *ab initio* models and then subjected to two rounds of 3D heterogeneous refinements. The resulting 1,104,007 particles were finally chosen for nonuniform refinement. The particles were then processed by focused 3D variability using a soft mask covering the binding protomer, where 428,882 particles were chosen for further processing with local CTF refinement and local refinement using cryoSPARC (63), resulting in a 2.89-Å-resolution cryo-EM structure (Table S3 and Fig. S5).

**Model building and refinement.** Model buildings for $MtrD_{CR103}$-CASP, $MtrD_{CR103}$-Col, and $MtrD_{CR103}$-LL were based on their corresponding cryo-EM maps. The structure of $MtrD_{CR103}$ was used as an initial model. Subsequent model rebuilding was performed using Coot (64). Structure refinements were performed using the phenix.real_space_refine program (65) from the PHENIX suite (66). The final atomic models were evaluated using MolProbity (67). The statistics associated with data collection, 3D reconstruction, and model refinement are included in Table S3.

**Growth and CASP treatment of *N. gonorrhoeae* and *N. subflava*.** *N. gonorrhoeae* strain FA1090 (or strain FA19 or CDC2) cells were grown overnight at 37°C under a 5% (vol/vol) $CO_2$-enriched atmosphere on GCB agar (gonococcal base) plates containing Kellogg's supplements I and II (68). For growth in liquid medium, $10^7$ CFU from cultures grown overnight on GCB agar plates were resuspended in 100 $\mu$L GC broth containing Kellogg's supplements I and II and 0.042% (wt/vol) sodium bicarbonate in 96-well plates. A total of $10^7$ CFU of gonococcal strain FA1090 was seeded into 100 $\mu$L of GCB broth alone and GC broth containing Nov at sublethal levels (0.06 or 0.125 $\mu$g/mL), with or without 2.0 mg/mL of CASP. Plates were incubated statically at 37°C in a $CO_2$ incubator, and the $OD_{600}$ was read every hour using a VictorX3 2030 multilabel reader (PerkinElmer).

*Neisseria subflava* (ATCC 49275) cells were grown in GCB broth containing 0.042% sodium bicarbonate at 37°C. Cells were harvested at an $OD_{600}$ of 0.6, washed with GCB medium once, resuspended in GCB medium to an $OD_{600}$ of 1.0, and then further diluted to an $OD_{600}$ of 0.1 with GCB medium supplemented with 0.5 $\mu$g/mL novobiocin and 0.042% sodium bicarbonate. Cells were inoculated into a 96-well plate with a total volume of 100 $\mu$L. A range of final concentrations (0 to 2.0 mg/mL) of the CASP peptide with or without a sublethal level of Nov was added to each well. The experiments were performed at 37°C on a microplate reader (BioTek). The $OD_{600}$ value was monitored and recorded every 30 min within a 24-h period. Each experiment was repeated a minimum of three times.

**Molecular modeling.** The AutoDock Vina program (50) was used to predict CASP binding modes in *N. gonorrhoeae* $MtrD_{CR103}$, *E. coli* AcrB, and *P. aeruginosa* MexB. The binding protomers of $MtrD_{CR103}$ (from the $MtrD_{CR103}$-CASP structure from this study), AcrB (PDB accession number 2DRD) (45), and MexB (PDB accession number 2V50) (69) were used, where the bound ligands were removed before computational calculations. For each calculation, the protein was set as a rigid structure, whereas the conformation of the CASP molecule was optimized via all modeling and docking procedures. The grid size was set to 26 by 30 by 26 *xyz* points with grid spacing of 1 Å, and the grid center was assigned at an *x* value of 139.783, a *y* value of 145.195, and a *z* value of 151.514.

**Growth and CASP treatment of Gram-negative bacilli.** *E. coli* K-12 Δ*tolC::kan* cells were grown in LB medium containing 25 $\mu$g/mL kanamycin at 37°C to an $OD_{600}$ of 0.6. Cells were collected, washed with LB three times to remove the antibiotic kanamycin, resuspend in LB to an $OD_{600}$ of 1.0., and then further diluted to an $OD_{600}$ of 0.1 with LB supplemented with 2 $\mu$g/mL novobiocin. Cells were inoculated into a 96-well plate with a total volume of 100 $\mu$L. A final concentration of 0, 0.1, 0.2, 0.5, 1.0, 1.5, or 2.0 mg/mL of the CASP peptide was added to each well. The experiments were performed at 37°C on a microplate reader (BioTek). The $OD_{600}$ value was monitored and recorded every 30 min within a 24-h period. Each experiment was repeated a minimum of three times.

*E. coli* K-12 (BW25113) cells were grown in LB medium at 37°C to an $OD_{600}$ of 0.6. Cells were collected, washed with LB once, resuspend with LB to an $OD_{600}$ of 1.0, and then further diluted to an $OD_{600}$ of 0.1 with LB supplemented with 75 $\mu$g/mL Nov. Cells were inoculated into a 96-well plate with a total volume of 100 $\mu$L. A final concentration of 0, 0.1, 0.2, 0.5, 1.0, 1.5, or 2.0 mg/mL of the CASP peptide was added to each well. The experiments were performed at 37°C on a microplate reader (BioTek). The $OD_{600}$ value was monitored and recorded every 30 min within a 24-h period. Each experiment was repeated a minimum of three times.

*Salmonella enterica* subsp. *enterica* (ATCC 53648) cells were grown in LB medium at 37°C to an $OD_{600}$ of 0.6. Cells were collected, washed with LB once, resuspended with LB to an $OD_{600}$ of 1.0, and then further diluted to an $OD_{600}$ of 0.1 with LB supplemented with 128 $\mu$g/mL Nov. Cells were inoculated into a 96-well plate with a total volume of 100 $\mu$L. A final concentration of 0, 0.1, 0.2, 0.5, 1.0, 1.5, or 2.0 mg/mL of the CASP peptide was added to each well. The rest of the experimental procedures were the same as the ones described above for *E. coli* K-12 (BW25113) cells. Each experiment was repeated a minimum of three times.

*Pseudomonas aeruginosa* (ATCC 39327) cells were grown in LB medium at 37°C to an $OD_{600}$ of 0.6. Cells were collected, washed with LB once, resuspended with LB to an $OD_{600}$ of 1.0, and then further diluted to an $OD_{600}$ of 0.1 with LB supplemented with 512 $\mu$g/mL Nov. Cells were inoculated into a 96-well plate with a total volume of 100 $\mu$L. A final concentration of 0, 0.1, 0.2, 0.5, 1.0, 1.5, or 2.0 mg/mL of the CASP peptide was added to each well. The rest of the experimental procedures were the same as the ones described above for *E. coli* K-12 (BW25113) cells. Each experiment was repeated a minimum of three times.

**Data availability.** Atomic coordinates and EM maps of $MtrD_{CR103}$-CASP, $MtrD_{CR103}$-Col, and $MtrD_{CR103}$-LL have been deposited under PDB accession numbers 8DEU, 8DEV, and 8DEW and EMDB accession numbers EMD-27399, EMD-27400, and EMD-27401, respectively.

# SUPPLEMENTAL MATERIAL

Supplemental material is available online only.

**SUPPLEMENTAL FILE 1**, PDF file, 1.6 MB.

# ACKNOWLEDGMENTS

This work was supported by NIH grants R01AI145069 (E.W.Y.) and R01AI021150 (W.M.S.). W.M.S. is the recipient of a senior research career scientist award from the Biomedical Laboratory Research and Development Service of the U.S. Department of Veterans Affairs. This research was, in part, supported by the National Cancer Institute's National Cryo-EM Facility at the Frederick National Laboratory for Cancer Research under contract HSSN261200800001E. We thank Philip A. Klenotic for proofreading this manuscript.

The contents of this article are solely the responsibility of the authors and do not necessarily reflect the official views of the National Institutes of Health, the U.S. Department of Veterans Affairs, or the United States government.

We have no competing interest to declare.

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
