## [Reviewer comments · Microbiology Spectrum]

Microbiology Spectrum

Structural Basis of Peptide-Based Antimicrobial Inhibition of a Resistance-Nodulation-Cell Division Multidrug Efflux Pump

Meinan Lyu, Julio Ayala, Isabella Chirakos, Chih-Chia Su, William Shafer, and Edward Yu

Corresponding Author(s): Edward Yu, Case Western Reserve University

Review Timeline:

Submission Date:

August 1, 2022

Accepted:

August 23, 2022

Editor: Aixin Yan

Reviewer(s): Disclosure of reviewer identity is with reference to reviewer comments included in decision letter(s). The following individuals involved in review of your submission have agreed to reveal their identity: Hongmin Zhang (Reviewer #2)

Transaction Report:

DOI: <https://doi.org/10.1128/spectrum.02990-22>

August 23, 2022

Prof. Edward W Yu
Case Western Reserve University
Cleveland

Re: Spectrum02990-22 (Structural Basis of Peptide-Based Antimicrobial Inhibition of a Resistance-Nodulation-Cell Division Multidrug Efflux Pump)

Dear Prof. Yu:

I am pleased to let you know that your manuscript has been accepted, and I am forwarding it to the ASM Journals Department for publication. You will be notified when your proofs are ready to be viewed.

Sincerely,

Aixin Yan
Editor, Microbiology Spectrum
